# Towards Better Propagation of Non-parametric GNNs

## Abstract

Recent works show great interest in designing Graph Neural Networks (GNNs) that scale to large graphs. While previous work focuses on designing advanced sampling techniques for existing GNNs, the design of non-parametric GNNs, an orthogonal direction for scalable performance, has aroused lots of concerns recently. For example, nearly all top solutions in the Open Graph Benchmark [1] leaderboard are non-parametric GNNs. Unlike most GNNs which alternately do feature propagation and non-linear transformation in each GNN layer, non-parametric GNNs execute the non-parametric propagation in advance and then feed the propagated features into simple and scalable models (e.g., Logistic Regression). Despite their high predictive performance and scalability, non-parametric GNNs still face two limitations. First, due to the propagation of over-smoothed features, they suffer from severe performance degradation along with the propagation depth. More importantly, they only consider the graph structure and ignore the feature influence during the non-parametric propagation, leading to sub-optimal propagated features. To address these limitations, we present non-parametric attention (NPA), a plug-and-play module that is compatible with non-parametric GNNs, to get scalable and deep GNNs simultaneously. Experimental results on six homophilic graphs and five heterophilic graphs demonstrate NPA enjoys high performance – achieves large performance gain over existing non-parametric GNNs, deeper architecture – improves non-parametric GNNs with large model depth, and high scalability – can support large-scale graphs with low time costs. Notably, it achieves state-of-the-art performance on the large ogbn-papers100M dataset.

## 1 Introduction

Graph Neural Networks (GNNs) have achieved great success in many graph-based applications in recent years, such as natural language processing (Liu & Wu, 2022; Wu et al., 2021), computer vision (Liu et al., 2021; Shi & Rajkumar, 2020), recommendation system (Wu et al., 2022; Gao et al., 2022), and drug discovery (Gaudelet et al., 2021; Deng et al., 2022). By recursively propagating the node embedding along edges, GNNs can enhance the representation of each node with its distant neighbors, thus improves the model performance. However, most existing GNNs have to repeatedly perform the computationally expensive and recursive feature propagation with the participation of the entire graph at each training epoch, leading to high computation costs in a single machine and high communication costs in distributed settings (Zhang et al., 2022a). Therefore, they can not scale well to large graphs, which hinders their popularity and development in many real scenarios.

An intuitive idea to tackle the scalability issue is to apply more advanced sampling techniques to existing GNNs, such as node-wise sampling (Hamilton et al., 2017; Chen et al., 2018a), layer-wise sampling (Chen et al., 2018b; Huang et al., 2018) and graph-wise sampling (Chiang et al., 2019; Zeng et al., 2020). Although these sampling techniques can alleviate the scalability issue to a certain extent, the computation and communication cost is still high due to the recursive propagation. Besides, the sampling quality highly influences the predictive performance of GNNs, and it is hard for these methods to tackle the trade-off between the training scalability and model performance.

Fortunately, some recent works attribute the success of GNNs to the non-parametric feature propagation rather than the parametric non-linear transformation. For example, as shown in SGC (Wu

---

[1]https://ogb.stanford.edu/docs/leader_nodeprop

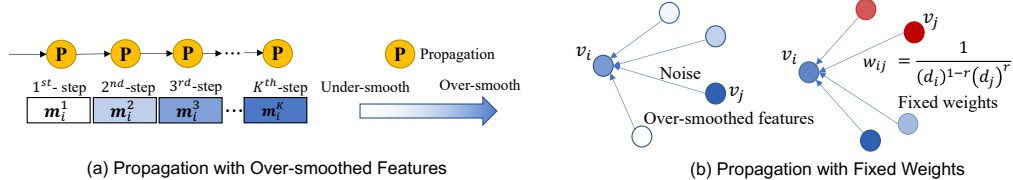

Figure 1: Example of two limitations in non-parametric GNNs.

et al., 2019), the non-parametric propagated features with a simple logistic regression model achieves comparable classification performance as GCN, while maintaining high scalability. As a new way to tackle the scalability issue, non-parametric GNNs have achieved lots of concern recently, and most top solutions in Open Graph Benchmark (Hu et al., 2020) are non-parametric GNNs. Following this direction, various non-parametric and scalable GNNs are proposed to further improve the predictive performance of SGC, such as SIGN (Frasca et al., 2020b), $S^2$GC (Zhu & Koniusz, 2021), GBP (Chen et al., 2020a) and NAFS (Zhang et al., 2022b). Despite their high scalability and predictive performance, existing non-parametric GNNs still face the following two limitations:

**Propagation with Over-smoothed Features.** Each node in a $K$-layer GNN can propagate its node feature $K$-steps to distant nodes, thus the node embedding can be smoothed and enhanced. However, as shown in Figure 1, such node information may be propagated to the full graph with a large propagation depth, making the node representation indistinguishable and leading to the bad predictive performance, i.e., the over-smoothing issue (Li et al., 2018). As discussed in Eq. 5, if we propagate the node feature infinite times in a connected graph, the final node embedding is only related to the node degrees. By combining the embedding of different propagation steps, some recent non-parametric GNNs can slightly alleviate the over-smoothing problem, and get better predictive performance with larger propagation depth. However, Figure 1 shows that the over-smoothed node embeddings (i.e., nodes with deep color) are continually propagated to their neighbors in the latter propagation process. Such redundant and noisy information will be propagated along the edges and harm the embedding of more nodes. To sum up, existing methods cannot fundamentally solve this over-smoothing issue due to the propagation of over-smoothed features.

**Propagation with Fixed Weights.** Existing non-parametric GNNs are built on the assumption of graph homogeneity, i.e., adjacent nodes are more likely to have the same label. However, the assumption does not hold in many real graph datasets. As shown in Figure 1 and introduced in Section 3.1, the propagation weight between two nodes in current non-parametric GNNs is only related to graph structures (e.g., the node degrees) and it is fixed in different propagation steps. Therefore, existing non-parametric GNNs may assign a large and fixed propagation weight to two nodes with different labels (i.e., nodes with different colors), making the embeddings of nodes with different labels indistinguishable. Besides the uncertain graph homogeneity, nodes with the same label may have a different extent of homogeneity (i.e., the label similarity), such as the nodes near and far from the decision boundary. Therefore, the weight should be adapted to their similarity rather than just the graph structure.

In this paper, we propose non-parametric attention (NPA), a plug-and-play module that is compatible with different non-parametric GNNs, to get scalable and deep GNNs simultaneously. As the over-smoothed features are global to capture the full graph information, they will not bring too much new information to current node embedding. Therefore, NPA introduces `global attention` to tackle the propagation of the over-smoothed features. Specifically, it assigns a node-adaptive attention weight to the newly propagated features to measure the newly generated global information compared with the previous feature. In this way, it can adaptively remove the over-smoothed and redundant feature information in each propagation step. Besides, as introduced in Appendix A.1, assigning larger weights to neighbors with more similar features leads to a more easy-to-learn classifier. To tackle the fixed propagation weights issue in the local feature propagation, NPA further proposes `local attention` to consider the similarity of propagated features when calculating the propagation weights and assigns more attention (i.e., larger propagation weight) to similar nodes adaptively in each propagation step. As introduced in Section 2.3, the key difference of existing non-parametric GNNs is how they combine different steps of propagated features. Since the goal of NPA is to improve the quality of different steps of propagated features, NPA is orthogonal and compatible with all kinds of existing non-parametric GNNs.

Our main contributions are as follows: (1) *New perspective*. To the best of our knowledge, this is the first work to explore the two limitations of existing non-parametric GNNs, i.e., the propagation with over-smoothed features and fixed weights; this new finding opens up a new direction towards scalable and deep GNNs. (2) *New method*. Based on our new findings, we propose NPA, a non-parametric attention method to tackle the above limitations. As a plug-and-play module, NPA is the first attempt to improve the propagation process of non-parametric GNNs, and it is compatible with existing non-parametric GNNs, to support scalable and deep GNNs simultaneously. (3) *High scalability and SOTA predictive performance*. Experimental results show that non-parametric GNNs equipped with NPA get large performance gains on five real-world datasets. Non-parametric GNNs with NPA can go deeper with better predictive performance. Specifically, NPA helps them to achieve the state-of-the-art performance on the large ogbn-papers100M dataset.

## 2 PRELIMINARY

### 2.1 NOTATIONS AND PROBLEM FORMULATION.

In this paper, we consider an undirected graph $\mathcal{G} = (\mathcal{V}, \mathcal{E})$ with $|\mathcal{V}| = n$ nodes and $|\mathcal{E}| = m$ edges. We denote $\mathbf{A}$ as the adjacency matrix of $\mathcal{G}$. Correspondingly, the degree matrix of $\mathbf{A}$ is denoted as $\mathbf{D} = \text{diag}(d_1, d_2, \cdots, d_n) \in \mathbb{R}^{n \times n}$, where $d_i = \sum_{v_j \in \mathcal{V}} \mathbf{A}_{ij}$. The node feature matrix is denoted as $\mathbf{X} = \{\boldsymbol{x}_1, \boldsymbol{x}_2, ..., \boldsymbol{x}_n\}$ in which $\boldsymbol{x}_i \in \mathbb{R}^f$ represents the feature vector of node $v_i$. We test the effectiveness of NPA in the widely used semi-supervised node classification task, and the goal is to predict the labels for nodes in the unlabeled set $\mathcal{V}_u$ with the supervision of labeled set $\mathcal{V}_l$.

### 2.2 GRAPH NEURAL NETWORKS

Based on the graph homogeneity assumption that locally connected nodes are likely to share the same label (McPherson et al., 2001), each node in most GNN models iteratively smooths the representations of its neighbors for better node embedding. The commonly used GNN layer updates the node feature embedding in the graph by propagating the features to their neighboring nodes:

$$\mathbf{X}^{(l)} = \delta\big(\hat{\mathbf{A}}\mathbf{X}^{(l-1)}\boldsymbol{\Theta}^{(l)}\big), \qquad \hat{\mathbf{A}} = \widetilde{\mathbf{D}}^{r-1}\tilde{\mathbf{A}}\widetilde{\mathbf{D}}^{-r}, \tag{1}$$

where $\widetilde{\mathbf{A}} = \mathbf{A} + \mathbf{I}_n$ is the adjacency matrix with self connection, $\mathbf{X}^{(l)}$ is the propagated node embedding matrix at layer $l$, $\mathbf{X}^{(0)}$ is the original node feature matrix, $\boldsymbol{\Theta}^{(l)}$ are the trainable weights, and $\delta$ is the activation function. $\hat{\mathbf{A}}$ is the normalized adjacency matrix, in which $\hat{\mathbf{A}}_{ij}$ represents how much information node $v_j$ should propagate to its neighboring node $v_i$. By setting $r = 0.5, 1$ and $0$, the normalized adjacency matrix $\widetilde{\mathbf{D}}^{r-1}\tilde{\mathbf{A}}\widetilde{\mathbf{D}}^{-r}$ represents the symmetric normalization adjacency matrix $\widetilde{\mathbf{D}}^{-1/2}\tilde{\mathbf{A}}\widetilde{\mathbf{D}}^{-1/2}$ (Klicpera et al., 2019), the transition probability matrix $\tilde{\mathbf{A}}\widetilde{\mathbf{D}}^{-1}$ (Zeng et al., 2020), and the reverse transition probability matrix $\widetilde{\mathbf{D}}^{-1}\tilde{\mathbf{A}}$ (Xu et al., 2018), respectively.

As shown in Eq. 1, Most GNN layer contains two operations: non-parametric feature propagation $\mathbf{P}$ and parametric feature transformation $\mathbf{T}$. Figure 2(a) shows the pipeline of parametric GNNs. For example, the $l$-th layer in GCN (Kipf & Welling, 2017) and GraphSAGE (Hamilton et al., 2017) firstly executes the non-parametric feature propagation (i.e., $\mathbf{P}$ operation) on the node embedding $\mathbf{X}^{(l-1)}$. Then, the propagated feature matrix $\widetilde{\mathbf{X}}^{(l-1)}$ is transformed with trainable weights $\boldsymbol{\Theta}^{(l)}$ and activation function $\delta$ (i.e., $\mathbf{T}$ operation) to generate new node embedding matrix $\mathbf{X}^{(l)}$.

### 2.3 NON-PARAMETRIC GRAPH NEURAL NETWORKS

To scale GNNs to large graphs, existing methods can be generally classified into the following two directions. One natural direction is to build more advanced sampling techniques (See details in Appendix A.14) for existing GNNs, and the other direction is to build non-parametric GNNs.

The main idea of non-parametric GNNs is to decouple the feature propagation and non-linear transformation in the GNN layer and finish the time-consuming feature propagation process without model parameter training. For each node $v_i$, non-parametric GNNs recursively propagates the original node feature $\mathbf{x}_i^0$ to its neighbors $K$ steps. The propagation process at the $t$-th step is as follows

$$\mathbf{m}_i^t \leftarrow \mathbf{P}\left(\{\mathbf{m}_j^{t-1}|j \in \mathcal{N}_i\}\right), \tag{2}$$

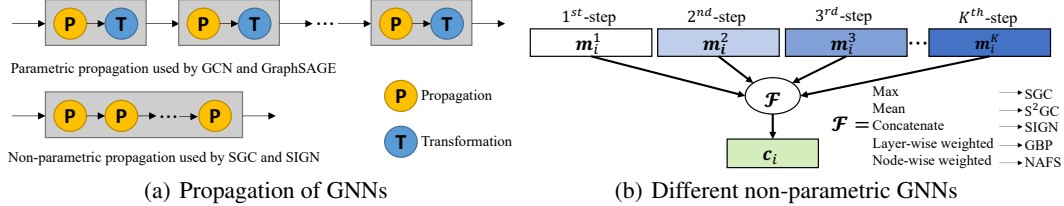

(a) Propagation of GNNs       (b) Different non-parametric GNNs

Figure 2: (Left) Example of parametric and non-parametric propagation. (Right) Illustration of different non-parametric GNNs.

where $\mathbf{m}_i^0 = \mathbf{x}_i$, and $\mathcal{N}_i$ are the neighbors of node $v_i$. After $K$-steps propagation, $\mathbf{m}_i^K$ at step $K$ can gather the neighborhood information from nodes that are $K$-hop away. The multi-step propagated features $\boldsymbol{\mathcal{M}}_i = \{\mathbf{m}_i^t \mid 0 \leq t \leq K\}$ are then combined into a single vector $\mathbf{c}_i$ as:

$$\mathbf{c}_i \leftarrow \boldsymbol{\mathcal{F}}(\mathcal{M}_i). \tag{3}$$

The key difference between existing non-parametric GNNs is the combination function $\boldsymbol{\mathcal{F}}$. Specifically, SGC (Wu et al., 2019) adopts the `Max` operation and feeds $\mathbf{m}_i^t$ to a simple logistic regression model for model training. Besides, SIGN (Frasca et al., 2020b) considers both the local and global information and adopts `Concatenate` operation on different steps of propagated features. Similarly, S²GC (Zhu & Koniusz, 2021) proposes to average the propagated features with the `Mean` operation. Besides, GBP (Chen et al., 2020a) assigns different weights to the propagated feature matrix and proposes the `Layer-wise weighted` operation to combine them adaptively. Last, motivated by the different smoothing speeds of nodes, NAFS (Zhang et al., 2022b) introduces the `Node-wise weighted` operation to combine the propagated features in a node-adaptive manner.

Similar to these works, we also use non-parametric GNNs for higher training scalability. The key difference lies in that we optimize non-parametric GNNs in an orthogonal way to the previous methods. Specifically, recent advancements in non-parametric GNNs aim at optimizing the combination function $\boldsymbol{\mathcal{F}}$. Different from them, our goal is to optimize the propagation method `P` to improve the quality of the multi-step propagated features $\boldsymbol{\mathcal{M}}_i = \{\mathbf{m}_i^t \mid 0 \leq t \leq K\}$. Therefore, the proposed NPA is compatible with the existing non-parametric GNNs and can be applied to them to further improve their predictive performance.

## 3 OBSERVATION AND INSIGHT

In this section, we make a deep analysis of the two limitations that exist in non-parametric GNNs and then provide some insights when designing our solution NPA.

### 3.1 OVER-SMOOTHED FEATURES

As stated by previous work (Wu et al., 2019; Zhu & Koniusz, 2021), the true success of GNNs lies in the non-parametric feature propagation rather than the parametric feature transformation. With a large propagation step, each node in non-parametric GNNs can propagate its feature information to distant neighbors, thus capture the deep graph structure information. Despite its benefits, a too-large propagation step may also lead to indistinguishable node embedding.

Concretely, if we execute $\hat{\mathbf{A}}\mathbf{X}$ for infinite times, the node embedding within the same connected component would reach a stationary state. When adopting $\hat{\mathbf{A}} = \widetilde{\mathbf{D}}^{r-1}\widetilde{\mathbf{A}}\widetilde{\mathbf{D}}^{-r}$, $\hat{\mathbf{A}}^\infty$ follows

$$\hat{\mathbf{A}}_{i,j}^\infty = \frac{(d_i + 1)^r (d_j + 1)^{1-r}}{2m + n}. \tag{4}$$

which shows that the influence from node $v_i$ to $v_j$ is only determined by their degrees. Then, the propagated feature matrix after infinite steps of propagation is

$$\mathbf{X}^\infty = \hat{\mathbf{A}}^\infty \mathbf{X}^0, \tag{5}$$

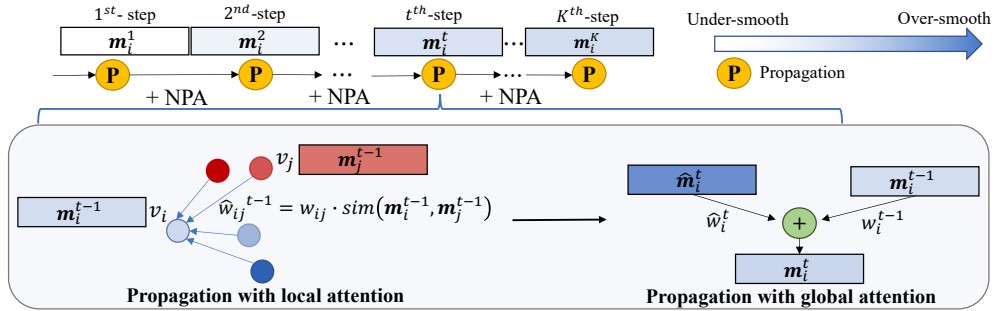

Figure 3: An overview of the proposed NPA.

where $\mathbf{X}^0$ is the original node feature matrix. Therefore, two nodes with the same graph structure information (e.g., the node degrees) will have the same propagated features no matter how different their original feature is. More details of the over-smoothing issue are explained in Appendix A.13.

Generally, the smoothing speed of nodes in different graph regions varies a lot. For example, as introduced in previous work (Zhang et al., 2022b), nodes with high degrees approach the stationary state more rapidly than the nodes with low degrees. Therefore, in the $t$-th propagation step, part of the nodes may already be over-smoothed while other nodes need further propagation due to their under-smooth states. The propagation of those over-smoothed features is in fact the propagation of feature noise, but this problem is under-explored in existing non-parametric GNNs.

## 3.2 FIXED PROPAGATION WEIGHTS

Most GNNs follow the graph homogeneity assumption that connected nodes are more likely to share the same label, thus they assign the propagation weights only based on the graph structure information.

Specifically, the $t$-th propagation step can be defined as

$$\mathbf{m}_i^t = \sum_{j \in \mathcal{N}_i} w_{ij} \mathbf{m}_j^{t-1}, \tag{6}$$

where $w_{ij}$ is the propagation weight, and it measures how much information we should propagate from node $v_j$ to $v_i$.

According to different types of normalized adjacency matrix (i.e., different values of $r$ in Eq. 1), the propagation weights of these methods can be defined as

$$w_{ij} = \frac{1}{(\tilde{d}_i)^{1-r}(\tilde{d}_j)^r}, \tag{7}$$

where $w_{ij} = \hat{\mathbf{A}}_{i,j}$ is the $i$-th row and $j$-th column of the smoothing matrix $\hat{\mathbf{A}}$, $\tilde{d}_i$ is the degree of node $v_i$ obtained from the adjacency matrix with self-connections $\widetilde{\mathbf{A}} = \mathbf{A} + \mathbf{I}_n$. Note that the propagation weights are fixed and applied in different steps of feature propagation.

Such fixed propagation weights may lead to low-quality node embedding in two perspectives. First, the graph homogeneity assumption cannot be guaranteed in many graph datasets, i.e., some connected nodes have different labels. As discussed in previous works (Zhu et al., 2020; Lim et al., 2021), the feature propagation of neighboring nodes with different labels is harmful to the node embedding. Besides, the homogeneity level for different nodes also varies (Du et al., 2022). For neighbors with high homogeneity, we should increase the propagation weights to make their embedding more similar. On the contrary, for neighbors with low homogeneity, we'd better decrease the propagation weights to avoid noise. As introduced in Appendix A.1, assigning larger weights to immediate neighbors with more similar features can lead to a more easy-to-learn classifier. Therefore, even for the graphs with good homogeneity property, existing non-parametric GNNs do not model them well enough because their propagation weights only consider the node degrees and ignore the feature similarity.

## 3.3 Design Insights of NPA

Despite the scalability of existing non-parametric GNNs, they suffer from the limitations of over-smoothed features and fixed propagation weights. Specifically, according to the observation in Sec. 3.1, some node embedding may already be over-smoothed due to their high smoothing speed. The over-smoothed features will not bring too much new information to current node embedding since they are global to capture the full graph information. Therefore, in different steps of feature propagation, we adopt `global attention` in NPA to node-adaptively measure the newly generated global information compared to existing ones and thus avoid the over-smoothed features. Besides, as introduced in Sec. 3.2, existing non-parametric GNNs adopt a fixed propagation weight and ignore the feature similarity during the local feature propagation. Therefore, we propose `local attention` in NPA to capture the similarity of propagated features in different steps of feature propagation, and thus make better use of the graph homogeneity.

## 4 Proposed Method

In this section, we present NPA, a non-parametric attention method that includes two attention mechanisms, i.e., `local attention` and `global attention`. Below, we first introduce an overview of NPA and then explain these two attention mechanisms in detail. At last, we summarize the advantages of the NPA over existing methods.

### 4.1 Method Overview

NPA is proposed to improve the propagation operation in non-parametric GNNs by tackling the issue of over-smoothed features and fixed weights. As shown in Figure 3, it contains two non-parametric attention mechanisms.

Specifically, for the $t$-th step feature propagation, NPA firstly introduces the feature similarity (i.e., $sim(\mathbf{m}_i^{t-1}, \mathbf{m}_j^{t-1})$) of node $v_i$ and $v_j$ to `local attention`. In this way, both the graph structure and node feature can be considered to better capture the graph homogeneity. After getting the propagated feature $\hat{\mathbf{m}}_i^t$ with the `local attention`, NPA measures the distance between $\hat{\mathbf{m}}_i^t$ and the existing one $\mathbf{m}_i^{t-1}$. For larger distance, NPA will assign $\hat{\mathbf{m}}_i^t$ a larger combination weight since it brings more new information to existing node embedding $\mathbf{m}_i^{t-1}$.

### 4.2 Local Attention

During the $t$-th feature propagation, we first measure the local similarity between the current node embedding $\mathbf{m}_i^{t-1}$ and its neighborhood embedding $\mathbf{m}_j^{t-1}$ and thus get $sim(\mathbf{m}_i^{t-1}, \mathbf{m}_j^{t-1})$. As discussed in Appendix A.1, the graph homogeneity is closely related to the feature similarity, and it is necessary to assign larger weights to more similar immediate neighbors. Therefore, we update the propagation weight $\hat{w}_{ij}^{t-1}$ as

$$\hat{w}_{ij}^{t-1} = w_{ij} sim(\mathbf{m}_i^{t-1}, \mathbf{m}_j^{t-1}), \tag{8}$$

where $sim(\cdot)$ is defined as the following RBF format

$$sim(\mathbf{m}_i^{t-1}, \mathbf{m}_j^{t-1}) = e^{-\frac{\left\| \mathbf{m}_i^{t-1} - \mathbf{m}_j^{t-1} \right\|_2}{2\sigma^2}}. \tag{9}$$

As introduced in Eq. 7, $w_{ij}$ is a fixed constant, while $sim(\mathbf{m}_i^{t-1}, \mathbf{m}_j^{t-1})$ varies in different steps of propagation, and thus make the propagation weight $\hat{w}_{ij}^{t-1}$ adaptive. We then use $\hat{w}_{ij}^{t-1}$ to propagate the node embedding and get the $t$-th propagated features $\hat{\mathbf{m}}_i^t$ as

$$\hat{\mathbf{m}}_i^t = \sum_{j \in \mathcal{N}_i} \hat{w}_{ij}^{t-1} \mathbf{m}_j^{t-1}. \tag{10}$$

 Larger $\hat{w}_{ij}^{t-1}$ means node $v_i$ pays more local attention to node $v_j$ during the propagation process, since these two nodes are more similar in both the graph structure and the $t-1$-step propagated features.

### 4.3 GLOBAL ATTENTION

Considering the propagation of over-smoothed features, we propose to update the node embedding and thus control the smoothing level after each step of propagation. Specifically, given the propagated node embedding $\hat{\mathbf{m}}_i^t$ and the embedding before propagation $\mathbf{m}_i^{t-1}$, we propose to measure how much new global information $\hat{\mathbf{m}}_i^t$ will introduce like $\hat{\alpha}_i^t = 1 - cos(\hat{\mathbf{m}}_i^t, \mathbf{m}_i^{t-1})$, where $cos(\cdot)$ is the cosine similarity function. Larger $\hat{\alpha}_i^t$ means the newly propagated node embedding $\hat{\mathbf{m}}_i^t$ is less similar to existing node embedding $\mathbf{m}_i^{t-1}$, and thus will introduce more benefits.

Therefore, we denote $\hat{\alpha}_i^t$ as the combination weight of $\hat{\mathbf{m}}_i^t$. Then, the combination weight for $\mathbf{m}_i^{t-1}$ is defined as $\alpha_i^{t-1} = 1 - \hat{\alpha}_i^t$. To better control the relative scale of these two combination weights, we normalize them with the temperature $T$ as

$$\hat{w}_i^t = \frac{e^{\hat{\alpha}_i^t/T}}{e^{\hat{\alpha}_i^t/T} + e^{\alpha_i^{t-1}/T}}, \quad w_i^{t-1} = \frac{e^{\alpha_i^{t-1}/T}}{e^{\hat{\alpha}_i^t/T} + e^{\alpha_i^{t-1}/T}}. \tag{11}$$

At last, we weighted combine the propagated node feature $\hat{\mathbf{m}}_i^t$ and existing node feature $\mathbf{m}_i^{t-1}$, and get the final $t$-th step node embedding $\hat{w}_i^t$ as

$$\mathbf{m}_i^t = \hat{w}_i^t \hat{\mathbf{m}}_i^t + w_i^{t-1} \mathbf{m}_i^{t-1}. \tag{12}$$

Generally, the over-smoothed features are too global and they introduce fewer benefits to existing node embedding. With this `global attention` mechanism, NPA can control the smoothing level of $\mathbf{m}_i^t$ in an adaptive manner, and thus avoid the over-smoothing issue.

### 4.4 NOVELTY AND CHARACTERISTICS OF NPA

**Novelty of Local Attention.** The main novelty of `local attention` is its motivation (the under-explored new limitation), i.e., the propagation with fixed weights in non-parametric GNNs. To tackle this limitation, we simplify the parametric attention mechanism in (Veličković et al., 2017; Zhang et al., 2018; Brody et al., 2022) to non-parametric local attention to maintain the high scalability of non-parametric GNNs, and this is the first attempt to improve the propagation process in non-parametric GNNs.

**Novelty of Global Attention.** Similarly, using adaptive weights in non-parametric GNNs is common, such as GBP, NAFS, and (Chien et al., 2021; He et al., 2022). However, as discussed in Eq. 3, existing non-parametric GNNs aim at optimizing the combination function $\mathcal{F}$ with adaptive weights. Different from them, we adopt global attention to tackle the propagation of over-smoothing features in the propagation process. And we aim to optimize the propagation method $\mathbf{P}$ (defined in Eq. 2) to improve the quality of the multi-step propagated features, rather than the combination function $\mathcal{F}$ in existing non-parametric GNNs.

More comparisons to weighted residual connection, NAFS and GAT are summarized in Appendix A.10. Besides, we have further analyzed the advantages (i.e., high scalability, high effectiveness and high compatibility) of NPA over existing methods in Appendix A.11.

## 5 EXPERIMENTS AND RESULTS

To evaluate the effectiveness of the proposed NPA module, we plug it into five typical non-parametric GNNs and conduct extensive experiments on real-world datasets. Specifically, we aim to answer the following questions: **Q1**: Does the performance of existing non-parametric GNNs get improved when plugging NPA in? **Q2**: Do the `local attention` and the `global attention` in NPA really help? **Q3**: Can NPA be used in both the homophilic and heterophilic graphs? **Q4**: Does NPA help existing non-parametric GNNs toward deeper? **Q5**: Does NPA help existing non-parametric GNNs perform better on different sparse scenarios? **Q6**: How does NPA affect the feature propagation time, end-to-end training time and converge time in practical?

### 5.1 EXPERIMENTS SETUP

**Datasets.** We conduct the experiments on 1) three homophilic citation network datasets (i.e., Cora, Citeseer, and PubMed) (Kipf & Welling, 2017), 2) three homophilic OGB datasets (i.e., ogbn-arxiv,

Table 1: Node classification performance on homophilic graphs.

| Method | Citation datasets | | | OGB datasets | | |
|---|---|---|---|---|---|---|
| | **Cora** | **Citeseer** | **PubMed** | **ogbn-arxiv** | **ogbn-products** | **ogbn-papers100M** |
| SGC | 81.0±0.0 | 71.9±0.1 | 78.9±0.0 | 70.78±0.22 | 74.08±0.15 | 63.29±0.19 |
| SGC+NPA | **83.0±0.0** | **73.6±0.0** | **80.1±0.0** | **71.64±0.29** | **75.32±0.17** | **64.82±0.42** |
| S$^2$GC | 83.5±0.0 | 73.6±0.1 | 80.2±0.0 | 72.01±0.25 | 76.84±0.20 | 65.03±0.35 |
| S$^2$GC+NPA | **83.7±0.1** | **74.0±0.0** | **80.8±0.0** | **72.31±0.21** | **77.01±0.12** | **65.42±0.12** |
| SIGN | 82.1±0.3 | 72.4±0.8 | 79.5±0.5 | 71.93±0.30 | 76.83±0.39 | 64.28±0.14 |
| SIGN+NPA | **83.1±0.5** | **73.8±0.8** | **80.2±0.6** | **72.39±0.21** | **77.58±0.06** | **65.40±0.14** |
| GBP | 83.9±0.7 | 72.9±0.5 | 80.6±0.4 | 71.97±0.20 | 76.71±0.12 | 64.76±0.51 |
| GBP+NPA | **84.1±0.7** | **73.4±0.4** | **80.7±0.4** | **72.19±0.16** | **76.93±0.17** | **65.48±0.29** |
| NAFS | 84.1±0.6 | 73.5±0.4 | 80.3±0.4 | 71.10±0.62 | 75.27±0.21 | 64.70±0.40 |
| NAFS+NPA | **84.2±0.6** | **73.8±0.5** | **80.9±0.3** | **71.65±0.31** | **75.44±0.17** | **65.04±0.09** |

Table 2: SOTA performance on ogbn-papers100M

| Method | GAMLP | GAMLP+NPA | GLEM+GIANT+GAMLP | GLEM+GIANT+GAMLP+NPA |
|---|---|---|---|---|
| Accuracy | 67.71± 0.2 | 67.83± 0.11 | 70.37± 0.02 | 70.44± 0.04 |

ogbn-products, ogbn-papers100M) (Hu et al., 2020), and 3) five heterophilic datasets (i.e., Texas, Wisconsin, Cornell, Film, and ogbn-mag) (Pei et al., 2020; Hu et al., 2020). For all datasets, we adopt the official training/validation/test split. More details about these datasets are in Appendix A.2.

**Baselines.** We plug the proposed NPA module into six typical non-parametric GNNs and compare them with their original versions, respectively. Following the taxonomy of the combination function $\mathcal{F}$ in Eq. 3, we select 1) SGC (Wu et al., 2019) as a representative of `Max` operation, 2) SIGN (Frasca et al., 2020a) as a representative of `Concatenate` operation, 3) S$^2$GC (Zhu & Koniusz, 2021) as a representative of `Mean` operation, 4) GBP (Chen et al., 2020a) as a representative of `Layer-wise weighted` operation, 5) NAFS (Zhang et al., 2022b) as a representative of `Node-wise weighted` operation, and 6) GAMLP (Zhang et al., 2022c) as a representative of `Node-wise weighted` operation, and the current SOTA model in ogbn-papers100M.

**Implementations.** To alleviate the influence of randomness, we repeat each method ten times and report the mean predictive accuracy and the corresponding standard deviation. The hyper-parameters of each method are tuned with random search or set according to the original paper if available. Please refer to Appendix A.2 for more details. Our code is available in the anonymized repository `https://anonymous.4open.science/r/NPA`.

## 5.2 EXPERIMENTAL RESULTS

**End-to-end Performance comparison on homophilic graphs.** To answer **Q1** and verify the performance improvement of NPA, we conduct node classification tasks on six homophilic graph datasets. Table 1 shows the test accuracy of the baselines with and without the proposed NPA module. Experimental results show that the proposed NPA consistently help all baseline methods achieve better predictive performance on three citation datasets and three OGB datasets. Since each baseline method represents one kind of the combination function $\mathcal{F}$, the proposed NPA can plug in different non-parametric GNNs and obtain performance improvements without losing scalability.

**SOTA performance in ogbn-papers100M.** To test the performance of NPA in extremely large graph, we equip GLEM+GIANT+GAMLP with NPA, and the experimental results in Table 5 show that NPA helps the non-parametric GNN model (i.e., GAMLP) to get the new SOTA results in the large ogbn-papers100M dataset. Note that both GIANT (Chien et al., 2022) and GLEM (Zhao et al., 2022) aim at improving the raw feature of OGB, and GAMLP is currently the SOTA GNN model in this dataset, more concrete analysis can be found in Appendix A.3

**Ablation study.** To answer **Q2**, we conduct experiments with different GNNs on three citation datasets to validate the effectiveness of the `local attention` and the `global attention`. And the results show that both these two attention mechanisms are necessary and co-contribute to a well-performed GNN model. More experimental results and analysis are in Appendix A.4. And we also give interpretability analysis of these two attention mechanisms in Appendix A.9.

Table 3: Node classification performance on heterophilic graphs. We define homophily as the fraction of edges in a graph whose endpoints have the same label (Yu et al., 2022).

| Method homophily | Texas 0.04 | Wisconsin 0.11 | Cornell 0.18 | Film 0.17 | ogbn-mag 0.30 |
|---|---|---|---|---|---|
| SGC | $57.30 \pm 8.18$ | $50.59 \pm 4.62$ | $59.73 \pm 5.60$ | $27.17 \pm 1.23$ | $35.71 \pm 0.22$ |
| SGC+NPA | $\mathbf{80.27 \pm 5.55}$ | $\mathbf{83.73 \pm 3.04}$ | $\mathbf{80.00 \pm 5.57}$ | $\mathbf{36.28 \pm 0.93}$ | $\mathbf{37.08 \pm 0.28}$ |

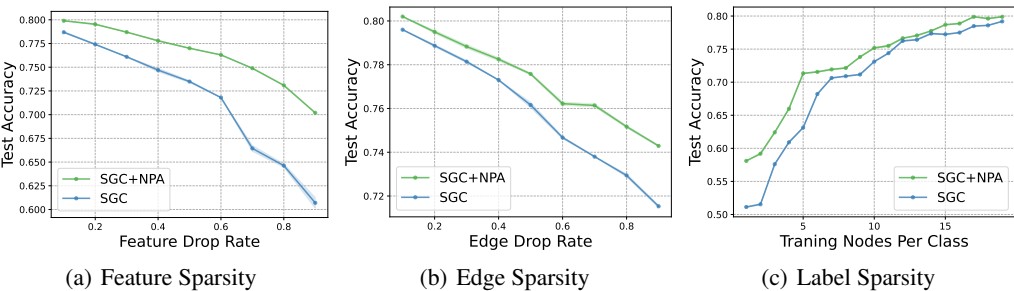

(a) Feature Sparsity  (b) Edge Sparsity  (c) Label Sparsity

Figure 4: Test accuracy on PubMed under different sparse scenarios(feature, edge, and label sparsity).

**End-to-end Performance comparison on heterophilic graphs.** To test the effectiveness of NPA on heterophilic graphs and answer **Q3**, we further conduct experiments on four heterophilic graphs. The experimental results in Table 3 show that SGC can achieve much better predictive performance on different heterophilic graphs with the help of NPA. The consistent performance improvement on these four heterophilic graphs comes from both `local attention` and `global attention`, which alleviate heterophily and over-smoothing problem, respectively. More details about the experimental analysis can be found in Appendix A.7.

**Towards Sparse Scenarios.** Many real-world graphs are sparse, and nodes in sparse graphs need deeper architecture to enhance their node embedding from distant neighbors. As NPA can help non-parametric GNNs toward deeper (see experimental results in Appendix A.5, answered **Q4**), we answer **Q5** and conduct experiments on the PubMed dataset under different sparse scenarios: feature sparsity, edge sparsity, and label sparsity. More details about the sparsity settings are in Appendix A.2. Results in Figure 4 show that with the help of NPA, the test accuracy of the non-parametric SGC model drops slower when the drop ratio gets larger and the number of training nodes gets fewer, demonstrating NPA can help non-parametric GNNs perform better on different sparse scenarios.

**Time cost.** According to the complexity and scalability analysis in Appendix A.11, NPA will not influence the scalability of existing non-parametric GNNs. But it indeed takes additional time to equip NPA. To answer **Q6**, we conduct exhaustive experiments to measure the time cost for feature propagation, end-to-end training, and training to converge under different settings. The experimental results show that NPA only introduces a little more time in practice and would maintain the high scalability of existing non-parametric GNNs. More details about the experimental results and analysis can be found in Appendix A.6.

## 6  CONCLUSION

This paper presents NPA, a non-parametric attention method towards scalable and deep GNNs. Unlike existing non-parametric GNNs which aim to improve the combination process of different steps of propagated features, NPA propose a new and orthogonal perspective to directly optimize the quality of propagated features. Specifically, considering the over-smoothed features and fixed weights in the propagation process of existing non-parametric GNNs, NPA proposes the `local attention` and `global attention` to tackle these two limitations respectively. As a plug-and-play module, NPA is compatible with all kinds of non-parametric GNNs, to improve their predictive performance while maintaining high training scalability. Experimental results on eleven real-world homophilic and heterophilic graph datasets demonstrate that NPA can consistently improve the classification accuracy of all non-parametric GNNs.

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

# A  APPENDIX

The appendix is organized as follows:

## A.1  THEORETICAL ANALYSIS

**Theorem 1.** *In discussed non-parametric GNNs, The classifier* $\Phi(\mathbf{x})$ *has 2 properties:*

- **property 1:** *Given feature* $\mathbf{x_a} \in R^f$ *(* $f$ *is feature dimension) and a non-zero scalar* $m \in R$, $\arg\max \Phi(\mathbf{x_a}) = \arg\max \Phi(m\mathbf{x_a})$.

- **property 2:** *Given two features* $\mathbf{x_a}, \mathbf{x_b} \in R^f$, *if* $\arg\max \Phi(\mathbf{x_a}) = \arg\max(\mathbf{x_b})$, *then* $\arg\max \Phi(\mathbf{x_a} + \mathbf{x_b}) = \arg\max \Phi(\mathbf{x_a}) = \arg\max \Phi(\mathbf{x_b})$.

**Proof.**  When $\Phi$ is a linear classifier, i.e., $\Phi(\mathbf{x}) = softmax(\mathbf{W}\mathbf{x})$, due to the property of linear transformation and softmax, Property 1 and Property 2 are held. When $\Phi$ is a Multi-Layer Perceptron, i.e., $\Phi(\mathbf{x}) = softmax(MLP(\mathbf{x}))$. Assume that the activation in MLP is ReLU. We decompose the proof into each layer in MLP. Each layer contains a linear transformation and ReLU. Since ReLU is monotonic increasing when $x \geq 0$, it will not distort the relative relationship of two scalars. Due to the property of linear transformation and ReLU, Property 1 and Property 2 are held in each hidden layer. In the output layer, softmax will not distort the relative relationship of two scalars, either. Thus Property 1 and Property 2 are also held in the final output.

We use $\mathbf{x_i}$ to denote the feature of node $v_i$ and use $y_i$ to denote the label of node $v_i$. Assume that the graph is ideally homogeneous, i.e., $\forall v_j \in N(v_i), y_j = y_i$. In other words, if we have a well-trained classifier $\Phi(\mathbf{x})$, we have:

$$\forall v_j \in N(v_i), \arg\max \Phi(\mathbf{x_i}) = \arg\max \Phi(\mathbf{x_j}). \tag{13}$$

Considering the node $v_i$ and its immediate neighbors $v_j \in N(v_i)$ and the corresponding $w_{ij}$, we assume that each neighbor's feature is close to the $x_i$ with error $\epsilon_{\mathbf{ij}}$:

$$\mathbf{x_j} = \mathbf{x_i} + \epsilon_{\mathbf{ij}}, \tag{14}$$

thus with weights $w_{ij}, \sum_{v_j \in N(v_i)} w_{ij} = 1$, the node's feature $x_i$ can be reconstructed by its immediate neighbors with errors:

$$\mathbf{x_i} = \sum_{v_j \in N(v_i)} w_{ij}\mathbf{x_j} - \sum_{v_j \in N(v_i)} w_{ij}\epsilon_{\mathbf{ij}}. \tag{15}$$

Here we sort the edge weights $w_{ij}$ in the descendant order, i.e., $\{w_{ij}\} = \{w_{ij_1} > w_{ij_2} > ... > w_{ij_{|N(v_i)|}}\}$. And rearrange its immediate neighbors' features such that the $L_2$ norm of $\epsilon_{ij}$ is ascendant, i.e., $\{\mathbf{x_j}\} = \{\mathbf{x_{j_1}}, ..., \mathbf{x_{j_k}}, \mathbf{x_{j_{k+1}}}, ..., \mathbf{x_{j_{|N(v_i)|}}}\}, \forall k \in [1, |N(v_i)| - 1], \|\epsilon_{\mathbf{ij_k}}\|_2 < \|\epsilon_{\mathbf{ij_{k+1}}}\|_2$, we have:

**Theorem 2.** *Assigning weight $w_{ij_k}$ to neighbor feature $\mathbf{x_{j_k}}$ in propagation can lead to learning a well-trained classifier easier.*

**Proof.** Since we assume that we have an ideal homogeneous graph, from the perspective of Eq. 13 of the homogeneous graph, Property 1 and Property 2 of well-trained classifier $\Phi$ in Theorem 1, we have:

$$
\begin{aligned}
\arg\max \Phi(\mathbf{x_i}) &= \arg\max \Phi(\mathbf{x_j}; v_j \in N(v_i)) \\
&= \arg\max \Phi(w_{ij}\mathbf{x_j}; v_j \in N(v_i)) \\
&= \arg\max \Phi(w_{ij}\mathbf{x_j} + w_{ik}\mathbf{x_k}; v_j, v_k \in N(v_i)) \\
&= \arg\max \Phi(\sum_{v_j \in N(v_i)} w_{ij}\mathbf{x_j}).
\end{aligned}
\tag{16}
$$

From another perspective, with Eq. 15, we have:

$$
\arg\max \Phi(\mathbf{x_i}) = \arg\max \Phi(\sum_{v_j \in N(v_i)} w_{ij}\mathbf{x_j} - \sum_{v_j \in N(v_i)} w_{ij}\epsilon_{\mathbf{ij}}).
\tag{17}
$$

Since both prediction $\arg\max \Phi(\sum_{v_j \in N(v_i)} w_{ij}\mathbf{x_j})$ and $\arg\max \Phi(\sum_{v_j \in N(v_i)} w_{ij}\mathbf{x_j} - \sum_{v_j \in N(v_i)} w_{ij}\epsilon_{\mathbf{ij}})$ are equal to the prediction $\arg\max \Phi(\mathbf{x_i})$, a expected classifier should make the same prediction given the two inputs. If the two inputs are closer, the classifier is much easier to train to make the same prediction. On the contrary, if the two inputs are far away, the classifier must approximate a complex decision manifold in the feature space, which could be intractable, leading to sub-optimal results. Thus, we measure the difference between $\sum_{v_j \in N(v_i)} w_{ij}\mathbf{x_j}$ and $\sum_{v_j \in N(v_i)} w_{ij}\mathbf{x_j} - \sum_{v_j \in N(v_i)} w_{ij}\epsilon_{\mathbf{ij}}$:

$$
\begin{aligned}
\textit{diff.} &= \|(\sum_{v_j \in N(v_i)} w_{ij}\mathbf{x_j}) - (\sum_{v_j \in N(v_i)} w_{ij}\mathbf{x_j} - \sum_{v_j \in N(v_i)} w_{ij}\epsilon_{\mathbf{ij}})\|_2 \\
&= \|\sum_{v_j \in N(v_i)} w_{ij}\epsilon_{\mathbf{ij}}\|_2 \\
&\leq \sum_{v_j \in N(v_i)} w_{ij}\|\epsilon_{\mathbf{ij}}\|_2,
\end{aligned}
\tag{18}
$$

and according to the *rearrangement inequality*, letting $\{w_{ij}\}$ be the reversed order of $\{\|\epsilon_{\mathbf{ij}}\|_2\}$ can minimize the upper bound of the difference, which may further reduce the difference between two inputs. To conclude, we showed that re-assigning propagation weights according to "how close between the neighbor node feature and its center node feature" is beneficial in achieving optimal prediction performance. From this perspective, our `local attention` is designed to additionally assign more weights to the nodes that have more similar features with that of their center nodes.

### A.2 EXPERIMENTAL DETAILS

**Dataset Description.** Cora, Citeseer, and PubMed[2] are three popular citation network datasets, and we follow the public training/validation/test split in GCN (Kipf & Welling, 2017). In these datasets, papers from different topics are considered nodes, and the edges are citations among the papers. The node attributes are binary word vectors, and class labels are the topics the papers belong to.

ogbn-arxiv is a directed graph, representing the citation network among all Computer Science (CS) arXiv papers indexed by MAG (Wang et al., 2020). ogbn-products is an undirected and unweighted graph, representing an Amazon product co-purchasing network. ogbn-papers100M is a directed citation graph of 111 million papers indexed by MAG. ogbn-mag dataset is a heterogeneous network composed of a subset of the MAG. It contains four types of entities: papers, authors, institutions, and fields of study, as well as four types of directed relations connecting two types of entities. Only paper node is associated with a 128-dimensional feature vector. The training/validation/test split in our

---

[2]https://github.com/tkipf/gcn/tree/master/gcn/data

Table 4: Overview of the 11 Datasets

| Dataset | #Nodes | #Features | #Edges | #Classes |
|---|---|---|---|---|
| Cora | 2,708 | 1,433 | 5,429 | 7 |
| Citeseer | 3,327 | 3,703 | 4,732 | 6 |
| PubMed | 19,717 | 500 | 44,338 | 3 |
| ogbn-arxiv | 169,343 | 128 | 1,166,243 | 40 |
| ogbn-products | 2,449,029 | 100 | 61,859,140 | 47 |
| ogbn-papers100M | 111,059,956 | 128 | 1,615,685,872 | 172 |
| Texas | 183 | 1703 | 309 | 5 |
| Wisconsin | 251 | 1703 | 499 | 5 |
| Cornell | 183 | 1703 | 295 | 5 |
| Film | 7,600 | 931 | 33,544 | 5 |
| ogbn-mag | 1,939,743 | 128 | 21,111,007 | 349 |

experiment is the same as the public version. The public version provided by OGB[3] is used in our paper.

Texas, Wisconsin, and Cornell are three heterophilic datasets of WebKB (Craven et al., 1998), which is a webpage dataset collected from computer science departments of various universities by Carnegie Mellon University. Nodes represent web pages, and edges are hyperlinks between them. Film is the actor-only induced subgraph of the film-director actor-writer network. The standard split protocol (Pei et al., 2020) is used in our paper.

The statistical analysis of these eleven graph datasets is summarized in Table 4.

**Hyper-parameters Setting.** In the proposed NPA, $\alpha$ in `local attention` and temperature $T$ in `global attention` are two additional hyper-parameters that need to be tuned. $\alpha$ controls the scale of $sim(\mathbf{m_i^{t-1}}, \mathbf{m_j^{t-1}})$ and determines how much it contribute to the updated weight $\hat{w}_{ij}$. Temperature $T$ controls the scale of the combination weights. We conduct coarse-grained pre-experiments to find the range of $\alpha$ in each baseline and dataset, and $T$ can simply be selected within [1,5,10,20,50]. We carefully tune the network depth for each method and dataset to get the best performance. Take the SGC method for an example, the depth we used for Cora, Citeseer, PubMed, ogbn-arxiv, ogbn-products and ogbn-papers100M is 10, 8, 14, 8, 13, 6, respectively.

As for the parametric classifiers, we categorize the classifiers of our baseline methods into two classes: Logistics Regression (LR) and Multi-Layer Perceptron (MLP). For LR, since it's sensitive to *weight decay*, we only tune *weight decay* and remain all other hyper-parameters following the settings in their original paper. For MLP, we only tune the *lr*, *weight decay* (if used in original papers), and *batch size* (if used in original papers) and remain all other hyper-parameters following the settings in their original paper.

**Sparsity Setting.** In feature sparsity, we randomly mask a portion (varying from 0.1 to 0.9) of node features. In edge sparsity, we randomly drop a portion (varying from 0.1 to 0.9) of the edges in the original graph. In both scenarios, the model needs to propagate more steps and pay more attention to the neighbors with high homogeneity to reconstruct reasonable node representations. In label sparsity, we enumerate the number of training nodes per class from 1 to 19. With less supervision signal, the classifier may not be well-trained, and the model needs to generate better node embedding in the non-parametric propagation process.

**Reproduction Instructions.** The experiments are conducted on a machine with Intel(R) Xeon(R) Gold 5120 CPU @ 2.20GHz, and a single NVIDIA TITAN RTX GPU with 24GB GPU memory. The operating system of the machine is Ubuntu 16.04. For software versions, we use Python 3.6, Pytorch 1.7.1, and CUDA 11.0. Our code is available in the anonymized repository `https://anonymous.4open.science/r/NPA`.

---

[3]https://ogb.stanford.edu/docs/nodeprop/

Table 5: sota performance on ogbn-papers100M

| Method | GAMLP | GAMLP+NPA | GLEM+GIANT+GAMLP | GLEM+GIANT+GAMLP+NPA |
|---|---|---|---|---|
| Accuracy | 67.71 | 67.83 | 70.37 | 70.44 |

Table 6: Ablation study on SGC and $S^2GC$

| Method | Cora | Citeseer | PubMed |
|---|---|---|---|
| SGC | 81.0±0.0 | 71.9±0.1 | 78.9±0.0 |
| SGC+NPA (w/o local attention) | 82.5±0.1 | 73.1±0.0 | 79.6±0.0 |
| SGC+NPA (w/o global attention) | 82.7±0.1 | 73.3±0.0 | 79.8±0.0 |
| SGC+NPA | **83.0±0.0** | **73.6±0.0** | **80.1±0.0** |
| $S^2GC$ | 83.5±0.0 | 73.6±0.1 | 80.2±0.0 |
| $S^2GC$+NPA (w/o local attention) | 83.6±0.1 | 73.7±0.1 | 80.3±0.1 |
| $S^2GC$+NPA (w/o global attention) | 83.6±0.1 | 73.9±0.1 | 80.6±0.1 |
| $S^2GC$+NPA | **83.7±0.1** | **74.0±0.0** | **80.8±0.0** |

## A.3 SOTA PERFORMANCE ON OGBN-PAPERS100M

As shown in current leaderboards[4], the SOTA model in ogbn-papers100M is GLEM+GIANT+GAMLP. GIANT is a node feature extraction method by self-supervised multi-scale neighborhood prediction, and GLEM takes raw node attributes and generates numerical node features with graph-structured self-supervision. To sum up, both GIANT and GLEM aim at improving the raw feature of OGB, and GAMLP is currently the SOTA GNN model in this dataset.

In fact, GAMLP is also one type of non-parametric GNN. Specifically, with recursive attention, GAMLP can tackle the over-smoothing issue by optimizing the combination function $\mathcal{F}$ with adaptive weights. However, GAMLP also faces two issues like other non-parametric GNNs: 1) Propagation with Over-smoothed Features; and 2) Propagation with Fixed Weights. So, we equip GLEM+GIANT+GAMLP with NPA and get the final results in Table 5.

The experimental results in Table 5 show that NPA can further improve the performance of both GAMLP and GLEM+GIANT+GAMLP. Notably, with the improved node features, GLEM+GIANT+GAMLP+NPA achieves the SOTA performance in ogbn-papers100M.

## A.4 ABLATION STUDY

We conduct an ablation study on three citation datasets to validate the effectiveness of the `local attention` and the `global attention`. We compare the SGC and $S^2GC$ equipped with NPA module (SGC+NPA and $S^2GC$+NPA) with three corresponding baselines respectively: 1) SGC/$S^2GC$+NPA (w/o local attention), which indicates that the propagation weights are fixed and only related to the graph structure, but `global attention` remains to control smoothing levels, 2) SGC/$S^2GC$+NPA (w/o global attention), which indicates that we put the over-smoothing issue aside, but `local attention` remains to adjust the weights of neighbors with different homogeneity, 3) SGC/$S^2GC$, indicating the original version of SGC/$S^2GC$.

Results in Table 6 show that the test accuracy of SGC/$S^2GC$+NPA (w/o local attention) and SGC/$S^2GC$+NPA (w/o global attention) improves on all datasets comparing the SGC/$S^2GC$, demonstrating the effectiveness of both `local attention` and `global attention`. Specifically, compared with SGC/$S^2GC$+NPA, the test accuracy of SGC/$S^2GC$+NPA (w/o local attention) drops more than that of SGC/$S^2GC$+NPA (w/o global attention), demonstrating that `local attention` has more benefit on predictive performance. However, `global attention` plays another important role: avoiding the over-smoothing issue and helping the model toward deeper. Thus, both `local attention` and `global attention` are necessary and co-contribute to a well-performed and deeper model.

---

[4]https://ogb.stanford.edu/docs/leader_nodeprop/#ogbn-papers100M

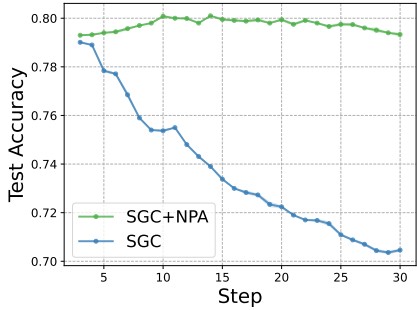

Figure 5: Test accuracy on PubMed along the propagation steps.

Table 7: Feature propagation time (second)

| Method | ogbn-arxiv | ogbn-products |
|---|---|---|
| SGC | 0.53 | 34.13 |
| SGC+NPA | 1.43 | 75.71 |
| SGC+NPA (w/o local attention) | 0.82 | 37.69 |
| SGC+NPA (w/o global attention) | 1.13 | 72.21 |

## A.5 PERFORMANCE WITH DEEPER ARCHITECTURE.

Existing non-parametric GNNs suffer from the propagation of the over-smoothed features when the propagation depth is large, resulting in the node representation indistinguishable and bad predictive performance. Thanks to the `global attention` in NPA, non-parametric GNNs equipped with NPA module can tackle this issue by assigning fewer combination weights to newly generated node embedding $\hat{\mathbf{m}}_i^t$ when little new information will be introduced. To test the performance of NPA with deeper GNNs, we select SGC as a baseline method and gradually enlarge the propagation depth on the PubMed dataset. Figure 5 shows that SGC suffers a lower test accuracy when the propagation depth gets larger, while SGC equipped with NPA (SGC+NPA) can always maintain a high test accuracy no matter how large the propagation depth is, demonstrating NPA can help non-parametric GNNs toward deeper.

## A.6 TIME COST ANALYSIS OF NPA

**Feature propagation time.** As discussed in Section A.11, both SGC and SGC+NPA have the same time complexity $\mathcal{O}(Kmf)$ in feature propagation process. But the calculation of the `local attention`($\mathcal{O}(Kmf)$) and `global attention`($\mathcal{O}(Knf)$) actually needs to cost additional time. To answer **Q6**, we firstly test the feature propagation time of 1) SGC, 2) SGC+NPA, 3) SGC+NPA (w/o local attention), and 4) SGC+NPA (w/o global attention) on two large datasets: ogbn-arxiv and ogbn-products. Results (in seconds) in Table 7 show that SGC+NPA costs more time than SGC, but generally costs no more than 3 times the time of SGC. Thus, the additional time cost of proposed NPA can be regarded as the time cost of SGC multiplied by a small constant factor. SGC+NPA (w/o local attention) costs less time than that of SGC+NPA (w/o global attention) since the time complexity of the `local attention` is usually larger than that of the `global attention`.

Table 8: End-to-End training time (preprocessed and trained on CPU)

| Method | ogbn-arxiv | ogbn-products |
|---|---|---|
| SGC | 155 | 551 |
| SGC+NPA | 156 | 593 |
| SGC+NPA (w/o local attention) | 155 | 555 |
| SGC+NPA (w/o global attention) | 155 | 589 |
| GCN | 587 | 14400 |
| GAT | 34800 | OOM |

Table 9: End-to-End training time (preprocessed on CPU and trained on GPU)

| Method | ogbn-arxiv | ogbn-products |
|---|---|---|
| SGC | 1.9 | 37 |
| SGC+NPA | 2.7 | 78 |
| SGC+NPA (w/o local attention) | 2.1 | 41 |
| SGC+NPA (w/o global attention) | 2.4 | 75 |
| GCN | 5.2 | 168 |
| GAT | 85 | OOM |

Table 10: Time for training to converge (preprocessed and trained on CPU)

| Method | ogbn-arxiv | ogbn-products |
|---|---|---|
| SGC | 1451 | 5130 |
| SGC+NPA | 1423 | 5102 |
| $S^2$GC | 1441 | 5102 |
| $S^2$GC+NPA | 1406 | 5095 |
| SIGN | 5508 | 48987 |
| SIGN+NPA | 5478 | 47602 |
| GBP | 1840 | 12700 |
| GBP+NPA | 1814 | 12705 |
| NAFS | 1401 | 5122 |
| NAFS+NPA | 1355 | 5100 |

**End-to-end training time.** We test the end-to-end training time (100 epochs, in seconds) of the 4 non-parametric GNNs: 1) SGC, 2) SGC+NPA, 3) SGC+NPA (w/o local attention), and 4) SGC+NPA (w/o global attention) and 2 parametric GNNs: 1) GCN, and 2) GAT on ogbn-arxiv and ogbn-products datasets. For the sake of 1) unified comparison of preprocessing time and training time and 2) demonstrating practical experiments, we conducted end-to-end training experiments on 1) pure CPU and 2) CPU when preprocessing and GPU when training, respectively. Results in Table 8 and Table 9 show that the parametric GNNs cost increasingly more time than non-parametric GNNs as the dataset gets larger. The difference between SGC and SGC+NPA is relatively small because SGC+NPA only needs to take a little more time in the feature propagation process and the following training process is the same with SGC.

**Time for training to converge.** We compare how long these methods take for training to converge with or without NPA. We conduct experiments on all 5 baseline methods. For SGC and SGC+NPA, $S^2$GC and $S^2$GC+NPA, NAFS and NAFS+NPA, since the NPA only has to do with the preprocessing phase and only the input feature maps are different, the time for training to converge each pair of methods take is similar. For GBP and GBP+NPA, following the original paper, the following MLPs are different from vanilla MLPs so they take relatively more time. For SIGN and SIGN+NPA, following the original paper, the following MLPs are larger so they take more time. The results (in seconds) are shown in Table 10 and Table 11.

## A.7 EXPERIMENTS ON HETEROPHILIC GRAPHS

The non-parametric GNNs baselines in this paper mostly make homophily assumptions, and thus perform badly on datasets with heterophily. Our proposed NPA, as a plug-and-play module, naturally

Table 11: Time for training to converge (preprocessed on CPU and trained on GPU)

| Method | ogbn-arxiv | ogbn-products |
|---|---|---|
| SGC | 7.4 | 31 |
| SGC+NPA | 7.0 | 29 |
| $S^2$GC | 6.8 | 29 |
| $S^2$GC+NPA | 6.6 | 27 |
| SIGN | 34 | 492 |
| SIGN+NPA | 31 | 481 |
| GBP | 24 | 578 |
| GBP+NPA | 24 | 570 |
| NAFS | 7.1 | 29 |
| NAFS+NPA | 7.0 | 27 |

Table 12: Node classification performance on heterophilic graphs. We define homophily as the fraction of edges in a graph whose endpoints have the same label (Yu et al., 2022).

| Method | Texas | Wisconsin | Cornell | Film | ogbn-mag |
|---|---|---|---|---|---|
| homophily | 0.04 | 0.11 | 0.18 | 0.17 | 0.30 |
| SGC | $57.30 \pm 8.18$ | $50.59 \pm 4.62$ | $59.73 \pm 5.60$ | $27.17 \pm 1.23$ | $35.71 \pm 0.22$ |
| SGC+NPA | $\mathbf{80.27 \pm 5.55}$ | $\mathbf{83.73 \pm 3.04}$ | $\mathbf{80.00 \pm 5.57}$ | $\mathbf{36.28 \pm 0.93}$ | $\mathbf{37.08 \pm 0.28}$ |

Table 13: Ablation study about normalization

| Method | Cora | Citeseer | PubMed |
|---|---|---|---|
| SGC | 81.0 | 71.9 | 78.9 |
| SGC+rbf (proposed) | 83.0 | 73.6 | 80.1 |
| SGC+rbf+normalization | 83.0 | 73.8 | 79.4 |

inherits their homophily assumption. However, NPA has the ability to improve the baseline methods in a heterophily setting without any other modification: the results in Table 12 show that SGC can achieve much better predictive performance on heterophilic graphs with the help of NPA.

Performance improvement on heterophilic graphs comes from both `local attention` and `global attention`, which alleviate heterophily and over-smoothing problem, respectively.

`local attention` in NPA helps baseline methods in two folds: First, it sets different weights on neighbors with the same label by feature similarity, aiming to select and consider the most "important" neighbors in propagation. Second, it reduces the propagation weights with respect to the neighbors with different labels, aiming to reduce the feature mix caused by those neighbors (which can be viewed as noise in homophily assumption). The second perspective is still working when we directly transfer the baseline methods with NPA in a heterophily setting. In `local attention` of NPA, the two connected nodes but with different labels may have quite different features (for example in the WebKB datasets different kinds of webpages contain different words), thus by calculating the feature similarity in the propagation phase, nodes will have a smaller propagation weight with respect to their neighbors with different labels. The node feature won't get mixed with its heterophilic neighbors' features, resulting in a low misclassification rate.

In `global attention`, the over-smoothing problem is solved at the feature level, just like the process in datasets with homophily.

In a word, the proposed NPA tackles both the heterophily and over-smoothing issue, making the baseline methods perform better.

## A.8  INSIGHTS IN MODEL DESIGNING

Here we offer some insights when choosing 1) with or without normalization in Eq. 10, 2) radial basis function in Eq. 10, 3) the value of $T$ in Eq. 11.

**About Normalization.**  Since we can use the $\sigma$ to control the similarity values, it can play a similar role with normalization: by adjusting $\sigma$, the weights in Eq. 10 can be controlled within a reasonable range (not too small or too large). We also test the SGC+rbf+normalization version on citation datasets (omit the std since the std is usually smaller than 0.1) and results are shown in Table 13. We find that the normalization after the radial basis function is not necessary but introduces additional time costs.

**Why choose radial basis function(rbf).**  We choose the radial basis function because we can use the hyper-parameter $\sigma$ to control the scale of the value of similarity, which can flexibly deal with different initial feature distributions. By adjusting $\sigma$, we can regulate the scale of the similarity values, which is instrumental in addressing varying initial feature distributions. In cases where initial features exhibit close proximity overall (even between disconnected nodes), a smaller $\sigma$ can be employed to discern similarity scores effectively. Conversely, when initial features are considerably far apart overall (even between connected nodes), a larger $\sigma$ can be used to guarantee that weights between two connected nodes (with the same label) are not unreasonably low. Besides, we need to balance the

Table 14: Comparison of radial basis function and cosine function

| Method | Cora | Citeseer | PubMed |
|---|---|---|---|
| SGC | 81.0 | 71.9 | 78.9 |
| SGC+rbf (proposed) | 83.0 | 73.6 | 80.1 |
| SGC+cos+normalization | 82.5 | 74.0 | 79.5 |

Table 15: Ablation study about $T$

| Method | Cora | Citeseer | PubMed |
|---|---|---|---|
| SGC | 81.0 | 71.9 | 78.9 |
| SGC+NPA | 83.0 | 73.6 | 80.1 |
| SGC+NPA (w/o $T$) | 82.8 | 73.6 | 79.9 |

structure and feature factors in propagation weights. Choosing a proper $\sigma$ can prevent the value from one factor from overwhelming that of the other one.

Another possible choice is the cosine function. But cosine does not have a parameter to control the value of similarity and thus has a drawback in its inflexibility to face different feature distributions. To prevent the cosine similarity value from squeezing in a range near 1 (when initial features are similar overall) or squeezing in a range near 0 (when initial features are not similar overall), normalization becomes necessary to control the value distinguishable and in a reasonable range. We also test this setting (SGC+cos+normalization) on citation datasets and results are shown in Table 14

**Value of $T$ in Eq. 11.** When regarding global attention, the motivation to use $T$ in Eq. 11 is to imitate the role of $\sigma$ in the radius basis function. We can use different $T$ to control the values in a proper range. In our experiments, choosing the value of $T$ is coarse: simply try some values: 1, 5, 10, 20, and 50 are enough. We report the ablation study about $T$ on the citation dataset in Table 15. We can see that introducing $T$ can benefit the performance of some datasets, which indicates that $T$ is indeed beneficial in a different distribution.

## A.9 INTERPRETABILITY ANALYSIS OF NPA

We here explain how the `local attention` and `global attention` work and why they are effective, respectively.

To explain how `local attention` works and why it helps, we define the *average homogeneous edge weight*, *average heterogeneous edge weight*, and *Edge Weight Margin* at step $t$ as

$$\textit{average homogeneous edge weight} = \frac{\sum_{i,j\in\mathcal{V},y_i=y_j}\hat{w}_{ij}^t}{\sum_{i,j\in\mathcal{V},y_i=y_j}1} \tag{19}$$

$$\textit{average heterogeneous edge weight} = \frac{\sum_{i,j\in\mathcal{V},y_i\neq y_j}\hat{w}_{ij}^t}{\sum_{i,j\in\mathcal{V},y_i\neq y_j}1} \tag{20}$$

$$\textit{Edge Weight Margin} = \frac{\textit{avg. homo. weight} - \textit{avg. hetero. weight}}{\textit{avg. hetero. weight}}. \tag{21}$$

*Edge Weight Margin* measures how much more attention we put on the neighbors with the same labels (which are called homogeneous neighbors) compared with those with different labels (which are called heterogeneous neighbors). Since homogeneity in the real-world graphs is uncertain, we hope the propagation can focus on the homogeneous neighbors and ignore the heterogeneous neighbors (which can be seen as noise).

Figure 6 shows that on the PubMed dataset, SGC+NPA which is equipped with `local attention` can enlarge *Edge Weight Margin* and put more attention on homogeneous neighbors by the measurement of local similarity. However, the original SGC uses fixed edge weights, which are only

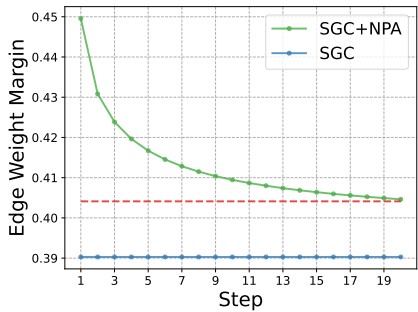

Figure 6: *Edge Weight Margin* on the PubMed dataset

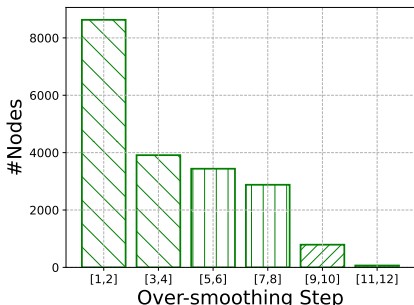

Figure 7: The over-smoothing step distribution (PubMed dataset)

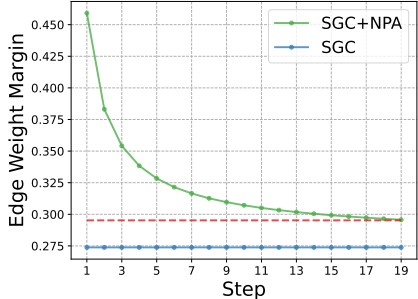

Figure 8: *Edge Weight Margin* on the Cora dataset

related to the graph structure, thus can not adjust the `local attention` flexibly, and takes on a low, straight blue line in Figure 6. Furthermore, an interesting observation in Figure 6 is that as the step gets larger, the *Edge Weight Margin* of SGC+NPA decreases since propagation will still take in a little part of the features of heterogeneous neighbors considering the graph structure, but the red curve will not intersect with the straight blue line standing for SGC. Rather, the curve tends to converge to a dashed green line above the SGC's line. We can observe a similar phenomenon in the Cora dataset (see Figure 8), which implies the generality of this observation.

To explain how `global attention` works and why it helps, we first put an observation that the steps which make node feature over-smoothed are varied. Following the previous work (Zhang et al., 2022b), we assume the node feature $\hat{\mathbf{m}}_i^t$ is over-smoothed if its combination weight $\hat{\alpha}_i^t$ is less than a small threshold $\epsilon$. By assigning $\epsilon = 0.05$, we define the over-smoothing step for node $i$ as

$$\text{over-smoothing step}(i) = \min\{t | \hat{\alpha}_i^t \leq 0.05\}, \tag{22}$$

and plot a histogram of the number of nodes with the over-smoothing step $t$ in Figure 7. Figure 7 shows that even with NPA, most node features get over-smoothed within 4 steps, while few node features won't get over-smoothed until propagating more than 8 steps. This observation corresponds to the two designs in `global attention`. First, since most node features will get over-smoothed within a few steps, it's necessary to use $\hat{\alpha}_i^t$ to measure how much new global information will be introduced. When the node features tend to be over-smoothed, the weights to combine newly propagated node embedding are much smaller, resulting in the avoidance of the over-smoothing issue. Second, since the over-smoothing steps of nodes can vary, we should measure the `global attention` node-wisely. At each step, nodes with different over-smoothed steps should have different combination weights. Thus, `global attention` helps avoid the over-smoothing issue and adapts to various over-smoothing steps of node features.

### A.10 THE NOVELTY AND CHARACTERISTICS OF NPA

**Compatibility with parametric GNNs.** In fact, both the `local attention` and `global attention` in NPA can also be compatible with parametric GNNs. Take the widely used GCN as an example, we can calculate the local attention based on the node embedding on the current layer, and then use the generated local attention to improve the adjacency matrix used in the propagation of its next layer. Besides, based on the similarity of previous node embedding and the newly propagated node embedding in the current layer, we can calculate the global attention and then use it to tackle the over-smoothing issue of the propagated node embedding. However, since parametric GNNs (e.g., GAT and DAGNN) can use parametric attention to achieve this goal, it may be unnecessary to introduce NPA.

**Novelty of Local Attention.** The main novelty of `local attention` is its motivation (the under-explored new limitation), i.e., the propagation with fixed weights in non-parametric GNNs. To tackle this limitation, we simplify the parametric attention mechanism in (Veličković et al., 2017; Zhang et al., 2018; Brody et al., 2022) to non-parametric local attention to maintain the high scalability of non-parametric GNNs, and this is the first attempt to improve the propagation process in non-parametric GNNs.

**Novelty of Global Attention.** Similarly, using adaptive weights in non-parametric GNNs is common, such as GBP, NAFS, and (Chien et al., 2021; He et al., 2022). However, as discussed in Eq. 3, existing non-parametric GNNs aim at optimizing the combination function $\mathcal{F}$ with adaptive weights. Different from them, we adopt global attention to tackle the propagation of over-smoothing features in the propagation process. And we aim to optimize the propagation method $\mathbf{P}$ (defined in Eq. 2) to improve the quality of the multi-step propagated features, rather than the combination function $\mathcal{F}$ in existing non-parametric GNNs.

Similar to `global attention`, another naive idea is the weighted residual connection in (Shen et al., 2016), which is used to train very deep networks. Weighted residual connection computes new features as below:

$$x_{i+1} = x_i + \lambda_i \Delta L_i, \tag{23}$$

where $\lambda_i$ is the weight scalar, $\Delta L_i$ is realized by two Conv-BN-ReLUs in the original paper, while here is realized by the propagation of features $Ax_i$. Original weighted residual connection set the $\lambda_i$ as learnable parameters optimized alone model training. However, when adopted in our scalable

Table 16: Comparison of `global attention` and weighted residual connection. Due to space limitations in the table, we use wrc as an abbreviation for weighted residual connection.

| Method | Cora | Citeseer | PubMed |
|---|---|---|---|
| SGC | 81.0 | 71.9 | 78.9 |
| SGC+wrc | 82.3 | 73.0 | 79.5 |
| SGC+local attention | 82.7 | 73.3 | 79.8 |
| SGC+local attention+wrc | 82.9 | 73.3 | 79.8 |
| SGC+NPA | 83.0 | 73.6 | 80.1 |
| $S^2$GC | 83.5 | 73.6 | 80.2 |
| $S^2$GC+wrc | 82.8 | 73.4 | 80.1 |
| $S^2$GC+local attention | 83.6 | 73.9 | 80.6 |
| $S^2$GC+local attention+wrc | 83.3 | 73.7 | 80.2 |
| $S^2$GC+NPA | 83.7 | 74.0 | 80.8 |

GNN setting, it cannot scale to large graphs since the propagation and training are not decoupled. To make the GNNs with weighted residual connection scalable, $\lambda_i$ can be treated as a hyperparameter and here we grid search it. However, we verify that it cannot beat our `global attention`, and is even not a right choice from the perspective of both experiments and theoretical analysis.

Table 16 shows the experimental comparison of our `global attention` and weighted residual connection. For SGC, the weighted residual connection can bring some benefits in predictive accuracy but can hardly make more improvement over `local attention`, while `global attention` (SGC+NPA) can make a further improvement over `local attention`. For $S^2$GC, the weighted residual connection even harms the performance of both $S^2$GC and $S^2$GC+local attention, let alone surpassing the performance achieved by `global attention` ($S^2$GC+NPA).

The reasons why weighted residual connection performs badly are two-fold: 1) whether $\lambda_i$ is learnable or not, it cannot adapt to each node. In one step, all nodes share the same weight, but our global attention can perform node-wisely to adapt to the truth that different nodes can have different smoothing speeds, which is also verified by experiments in Figure 7; 2) Whether $\lambda_i$ is learnable or not, the weighted residual connection is equivalent to simple propagation $x_{i+1} = Ax_i$ thus does not bring any new information or high-order feature cross. In weighted residual connection, any $x_i$ can be decomposed as $x_0 + \sum_1^i \lambda_i \Delta L_i$. Supposing a weighted combination function (e.g. $S^2$GC and GBP), by choose a series of weights $w_i$, the final features $\hat{x} = \sum_{i=0}^k w_i x_i = w_0 x_0 + \sum_{i=1}^k w_i \sum_{j=i}^k \lambda_i x_i = w_0 x_0 + \sum_{i=1}^k w_i (k-i+1)\lambda_i x_i$. However, this is equivalent to simple propagation $x_{i+1} = Ax_i$ since we can just choose a new series of weights $\hat{w}_i$ such that $\hat{w}_0 = w_0, \hat{w}_i = w_i(k-i+1)\lambda_i, i > 0$.

**Comparison with NAFS** (1) *Different motivation.* The goals of our method and NAFS are orthogonal. Similar to existing works (e.g., GBP and SIGN), NAFS highlights how to combine the propagated features, and it proposes a node-adaptive solution based on the Smoothing Weight. As discussed in Eq. 3, existing non-parametric GNNs aim at optimizing the combination function $\mathcal{F}$, but our goal is to optimize the propagation method $\mathbb{P}$ (defined in Eq. 2) to improve the quality of the multi-step propagated features.

(2) *Different design principles.* The global attention of our method is to control the node smoothing level at different propagation steps. Thus it adopts global attention to measure how much new information the propagated feature will introduce. Different from our method, the "Smoothing Weight" in NAFS only measures the distance between each node feature and its stationary state, without considering the over-smoothing issue during the feature propagation process. In this way, the over-smoothed feature will hurt the embedding quality of its neighbors in NAFS.

**Comparison with graph attention network (GAT).** (1) *New Motivation.* The main motivation of GAT is that the importance of different neighborhood nodes is different, thus it assigns different weights to adjacent nodes with the attention mechanism.

- Different from GAT, the main motivation of `local attention` in NPA is to consider the feature influence in the feature propagation since existing non-parametric GNNs follow the propagation process of SGC and only consider the graph structure information in feature propagation.

- Similar to existing non-parametric GNNs, GAT will also face the over-smoothing issue if we stack multiple GNN layers. Although some methods are proposed to tackle the over-smoothing issue by optimizing the combination function, the over-smoothed features will also be propagated to distant neighbors, thus decreasing the embedding quality.

(2) *New Solution.*

- Both the `local attention` in NPA and GAT adopts the attention mechanism to assign different weights to adjacent nodes, but the way they get the attention score is different. Specifically, GAT adopts a single-layer feedforward neural network parametrized by a weight vector, while NPA directly measures the similarity between two propagated node features. Besides, another difference is that our local attention can be pre-computed in the feature pre-processing, thus achieving higher scalability.

- To tackle this new motivation introduced above, the global attention is new to both non-parametric GNNs (e.g., GBP and NAFS) and parametric GNNs (e.g., GCN and GAT) since it aims to control the smoothing level of different layers of propagated features.

Specifically, as discussed in Eq. 3, existing non-parametric GNNs aim at optimizing the combination function $\mathcal{F}$ with adaptive weights. Different from them, we adopt global attention to tackle the propagation of over-smoothing features in the propagation process. And we aim to optimize the propagation method $\mathbf{P}$ (defined in Eq. 2) to improve the quality of the multi-step propagated features, rather than the combination function $\mathcal{F}$ in existing non-parametric GNNs.

## A.11 ADVANTAGES OVER EXISTING METHODS

**High Scalability.** NPA will not influence the scalability of existing non-parametric GNNs since it will not introduce any trainable parameters during the propagation process. NPA only needs to calculate the feature similarity of adjacent nodes, thus both SGC and SGC+NPA have the same time complexity $\mathcal{O}(Kmf)$, where $K$ is the maximum propagation step, $f$ is the feature dimension, and $m$ and $n$ are the number of edges and nodes respectively. Besides, NPA only requires storing the local and global attention weights, and the memory cost is $\mathcal{O}(m + n)$, which grows linearly with the edge number $m$ in typical real-world graphs. Due to the low memory cost and high efficiency, NPA is helpful to scale existing non-parametric GNNs to large graphs.

**High Effectiveness.** NPA improves non-parametric GNNs with non-parametric attention. Specifically, the `local attention` can help to better capture the graph homogeneity and the `global attention` enables non-parametric GNNs to capture deep structural information without the over-smoothing issue, making the non-parametric GNNs more effective.

**High Compatibility** Both the local and global attention introduced in NPA is proposed to improve the propagation process (defined in Eq. 2), rather than the combination process (defined in Eq. 3) considered by previous non-parametric GNNs. Therefore, NPA can benefit from recent advancements in non-parametric GNNs and be used as a plug-and-play module to further improve their performance.

## A.12 LIMITATIONS AND SOCIAL IMPACT

**Social Impact.** NPA can be employed in areas where graph modeling is the foremost choice, such as citation networks, social networks, chemical compounds, transaction graphs, road networks, etc. The effectiveness of NPA when improving the predictive performance and training scalability in those areas may bring a broad range of societal benefits. For example, accurately predicting the malicious accounts on transaction networks can help identify criminal behaviors such as stealing money and money laundering. Prediction on road networks can help avoid traffic overload and save people's time. A significant benefit of NPA is that it can support both scalable and deep graph learning.

**Limitations.** NPA faces the risk of information leakage in the smoothed features and attention weights. In this regard, we encourage researchers to understand the privacy concerns of NPA and investigate how to mitigate the possible information leakage.

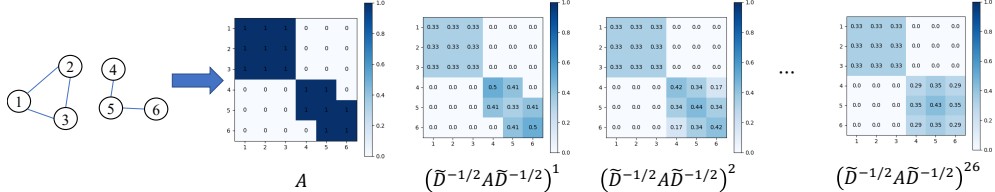

Figure 9: An example of the over-smoothing issue.

### A.13 EXAMPLE OF THE OVER-SMOOTHING ISSUE

To better clarify the over-smoothing issue during the feature propagation process of non-parametric GNNs. We set the normalized adjacency matrix as $\widetilde{\mathbf{D}}^{-1/2}\widetilde{\mathbf{A}}\widetilde{\mathbf{D}}^{-1/2}$, and then propagate it with an extremely large step (i.e., 26). As shown in Figure 9, no matter how different their original node features are, nodes with the degrees (e.g., node 4 and node 6) in a connected graph will have the same node feature after a large step of feature propagation. As a result, such indistinguishable node features will lead to the bad predictive performance of deep GNNs.

**The insight and comparison of NPA in overcoming over-smoothing.** There have been some works addressing over-smoothing issues. Similar to non-parametric GNNs, SHADOW-GNN (Zeng et al., 2021) can also scale to large graphs with a low-end GPU with limited memory capacity due to the minibatch computation, but it's time-consuming for model training is still high. Besides, similar to APPNP, GPRGNN (Chien et al., 2021) decouples the feature propagation and transformation. But they are hard to scale to large graphs since the time and memory-consuming feature propagation process can not be pre-processed as non-parametric GNNs. Finally, GCNII (Chen et al., 2020b) tackles the over-smoothing issue with the initial residual connection and identity mapping, but it also faces the scalability issue like the traditional GCN. Non-parametric GNNs can achieve the best tradeoff among memory consumption, training efficiency, and predictive performance.

The over-smoothing issue in non-parametric GNNs may exist in two cases. The former case is the final node embedding used for the downstream tasks, the latter is the node embedding during the feature propagation. A series of non-parametric GNNs (e.g., SIGN, GBP, and NAFS) have been proposed to address the over-smoothing issues for the former case, i.e., optimizing the combination function $\mathcal{F}$ (e.g., concatenate and weighted means) in Eq. 3. However, the issue of propagation with over-smoothed features (i.e., the latter case) has not been explored before in non-parametric GNNs. To the best of our knowledge, NPA is the first work that tackles this issue.

### A.14 DETAILS OF SAMPLING-BASED GNNS

An intuitive method to tackle the recursive neighborhood expansion problem in large-scale GNNs is sampling. And existing sampling techniques can be classified into three types: node-wise sampling, layer-wise sampling, and graph-wise sampling.

As a commonly used node-wise sampling method, GraphSAGE (Hamilton et al., 2017) samples a fixed-size set of neighbors in each propagation process. Besides, VR-GCN (Chen et al., 2018a) improves the node-wise sampling with variance reduction. In the layer level, Fast-GCN (Chen et al., 2018b) samples a fixed size of nodes at each layer, and ASGCN (Huang et al., 2018) adopts the adaptive layer-wise sampling with better variance control. For the graph-wise sampling, Cluster-GCN (Chiang et al., 2019) clusters the nodes and samples the nodes in the same clusters, and GraphSAINT (Zeng et al., 2020) directly samples a subgraph for model training.

Despite the success of these sampling-based GNNs, the sampling quality highly influences their predictive performance, and the computation and communication cost is still high in large graphs. Therefore, as an orthogonal way to tackle the scalability issue, our proposed method NPA is built on the more scalable non-parametric GNNs.

