# OpenReview forum: "Towards Better Propagation of Non-parametric GNNs"
_ICLR.cc/2024/Conference — ICLR 2024 Conference Withdrawn Submission_

### Official Review · Reviewer_zzGa · 2023-10-28

**Soundness:** 2 fair
**Presentation:** 3 good
**Contribution:** 1 poor
**Rating:** 3
**Confidence:** 5

**Summary:**

This paper improves the non-parametric GNN NAFS [1] by presenting a non-parametric pairwise similarity measurement. It consists of two components, local attention and global attention. The global attention is very similar to NAFS [1] with slight differences, while the novelty and contrition of local attention-based pairwise similarity are very limited. Experiments demonstrate its effectiveness.


[1] Wentao Zhang, Zeang Sheng, Mingyu Yang, Yang Li, Yu Shen, Zhi Yang, Bin Cui: NAFS: A Simple yet Tough-to-beat Baseline for Graph Representation Learning. ICML 2022: 26467-26483

**Strengths:**

1. The writing is easy to follow.
2. The motivation is clear.
3. The evaluations are basically sufficient to demonstrate its superiority.

**Weaknesses:**

1. My main concern is the novelty and contribution are very limited. Firstly, it is very similar to NAFS [1], especially the global components. Secondly, the local attention is very similar to GAT. The main difference is the non-parametric characteristic. However, this slight improvement is not enough to be accepted by ICLR.
2. The performance improvements are very limited compared to NAFS.

**Questions:**

See weakness.

---

### Official Review · Reviewer_whHX · 2023-10-30

**Soundness:** 3 good
**Presentation:** 3 good
**Contribution:** 2 fair
**Rating:** 5
**Confidence:** 5

**Summary:**

The paper proposes a novel Non-Parametric Attention (NPA) method for scalable and deep Graph Neural Networks (GNNs). Unlike existing non-parametric GNNs that improve feature propagation through a combination of different steps, NPA takes an orthogonal approach by directly optimizing the quality of feature propagation. The authors introduce local and global attention, which is somewhat similar to GAT, to address the issues of feature over-smoothing and fixed weight propagation in non-parametric GNNs. Experiments on eleven datasets validate the effectiveness of the proposed NPA compared with baselines.

**Strengths:**

1. The paper is easy to read, and generally well written.
2. The proposed method is well-motivated. The investigation of non-parametric GNNs is meaningful.
3. The experiments and ablations are abundant. Meanwhile, it analyzes the time cost.

**Weaknesses:**

- The novelty seems limited to some extent, which is my major concern. Roughly speaking, the idea of NPA is introducing the widely studied attention to SGC. The global attention is somewhat similar to GNNs with residuals, e.g., S2GC, APPNP, JKNet, etc. Although the designed local and global attention is not trainable (which is different from GAT), it is hard to regard the utilization of attention as a prominent contribution. The contributions may be not enough for ICLR.
- NPA fails to promote the non-linearity capacity of SGC/S2GC, which may be the primary limitation compared with vanilla GNNs. NPA just focuses on how to improve the graph propagation of non-parametric GNNs.
- As pointed out above, the achieved improvements, which are compared with non-parametric GNNs, are slight in most cases. It may be caused by the only improvement of graph propagation.

**Questions:**

Please see Weakness.

---

### Official Review · Reviewer_MrWb · 2023-10-31

**Soundness:** 2 fair
**Presentation:** 3 good
**Contribution:** 2 fair
**Rating:** 3
**Confidence:** 4

**Summary:**

In this paper, two attention techniques are introduced to address two key challenges in GNNs: the fixed propagation weight issue and the problem of over-smoothing. These attention scores are intuitively computed based on similarities. The motivation and explanation are clear presented with experimental validation.

**Strengths:**

1. The motivation behind this work is clear. It addresses two common issues in GNNs: the fixed propagation weight problem and the challenge of over-smoothing. To tackle the problem of fixed weights, the author introduces a technique called local attention, which assesses the similarity between node features. This method computes node-level similarity, and it is orthogonal to other pre-computing approaches.

2. To combat the over-smoothing problem, the author proposes the use of a residual connection, where the weights are also determined using similarity (global attention). The overall experiments conducted to validate these methods are presented and provide valuable insights.

**Weaknesses:**

1.	Overall, the novelty of this work is not strong. Given that fixed weight and over-smoothing problems have been extensively studied in the literature, there is no clear evidence of why and how the proposed local and global attention advances the state-of-the-art.

(1) Local attention: In the appendix, the authors assert that NPA directly quantifies the similarity between two propagated node features, setting it apart from GAT and the local attention mechanism can be pre-computed during feature pre-processing, which enhances scalability. However, it's still similar to the mechanisms in SAGN and GAMLP, as both of these models also involve precomputing attention based on propagated features. Combining the proposed method with those existing models might lead to only marginal improvements, as can be seen in the comparison between GAMLP and GAMLP+NPA in Table 2. It raises questions about the novelty of this technique.


(2) Global attention: The global attention mechanism appears to be primarily a residual connection with the previous aggregation step. The novelty lies in how the coefficients are computed using cosine similarity instead of being set manually. The novelty is limited. Besides, I also would suggest that the author should add an ablation study about how this approach addresses the over-smoothing issue in comparison to simpler techniques like APPNP and DAGNN (Towards Deeper Graph Neural Networks, KDD 2020).

2.	The Baselines are mostly weak in handling over-smoothing issues. Improvements over existing solutions need to be better justified.

(1) The experimental results indicate that SIGN+NPA performs worse than SAGN on two widely used large-scale datasets, ogbn-products, and ogbn-papers100M, where SAGN is a widely cited method whose technique is quite similar to this paper. In light of this, it is recommended that the author considers introducing more strong baselines and conducting additional experiments, such as SAGN/SAGN+NPA, to ascertain whether local attention indeed provides substantial benefits.

(2) Some of the baseline results might not be accurate. For instance, SIGN achieves an 80.5% score on the ogb-leaderboard, which is notably higher than the reported figures (76.83%) in the paper and considerably surpasses SIGN+NPA. It raises the question of whether the author employed fair experimental settings across these models.

3. Empirically, the improvements over weak baselines are pretty marginal (refer to Table 1 and Table 2) while the computation cost is higher. Moreover, there is no theoretical analysis to justify the advantages of the proposed method over existing ones.

4. The name "Non-parametric GNNs" in the title and paper is quite misleading. It essentially means the decoupled propagation similar to SGC, SIGN, and APPNP. It would be better to be adjusted to avoid confusion with terminologies in non-parametric statistics.

**Questions:**

Please refer to the weakness.

---

### Official Review · Reviewer_DswV · 2023-11-01

**Soundness:** 2 fair
**Presentation:** 3 good
**Contribution:** 2 fair
**Rating:** 3
**Confidence:** 5

**Summary:**

This paper introduces the attention mechanism into the propagation of the decoupled GNNs. Specifically, the authors propose both local attention to adjust the local propagation weight and global attention to control the residual weight. Experimental results demonstrate that the proposed method can be integrated into different decoupled GNNs and improve performance.

**Strengths:**

1. The proposed method can be integrated into different decoupled GNNs
2. The paper is well-written and easy to follow.

**Weaknesses:**

1. The authors claim that there are two limitations of the current decoupled GNNs. However, it is already solved by the existing method, or the proposed method can't solve the issue.
(a) The propagation will lead over-smoothed features. This problem can be solved by the residule connection such as APPNP.  Besides, is there any theoretical analysis to demonstrate that the proposed method can solve the oversmoothing issue?
(b) Existing non-parametric GNNs are built on the assumption of graph homogeneity. Despite the fact that the proposed method introduces attention, it is still based on the graph homogeneity assumption.
2. The original feature may not measure the similarity between nodes. Two nearby nodes with the same label may have different original features. Can the authors conduct experiments on non-citation datasets, such as the Amazon datasets?
3. The improvement of the proposed method is marginal.

**Questions:**

Please refere to the weakness.